# Treating Adenovirus Infection in Transplant Populations: Therapeutic Options Beyond Cidofovir?

**DOI:** 10.3390/v17050599

**Published:** 2025-04-23

**Authors:** Niyati Narsana, David Ha, Dora Y. Ho

**Affiliations:** 1Division of Infectious Diseases, UC Davis Medical Center, Sacramento, CA 95817, USA; 2Stanford Antimicrobial Safety and Sustainability Program, Stanford Health Care, Stanford, CA 94305, USA; 3Division of Infectious Diseases & Geographic Medicine, Stanford University School of Medicine, Stanford, CA 94305, USA

**Keywords:** adenovirus, adoptive T-cell therapy, brincidofovir, cidofovir, ganciclovir, ribavirin, hematopoietic stem cell transplant, solid organ transplant

## Abstract

Adenovirus (AdV) infections can lead to significant morbidity and increased mortality in immunocompromised populations such as hematopoietic stem cell and solid organ transplant recipients. This review evaluates currently available and emerging therapies for AdV infections. Cidofovir, while most commonly used, is limited by its variable efficacy and nephrotoxicity. This led to the development of brincidofovir, which has a better safety profile and great in vitro potency against AdV. The use of ribavirin and ganciclovir has been reported in the literature, but their use is limited due to inconsistent efficacy. Immune-based approaches, such as adoptive T-cell therapy, have shown promise in achieving viral clearance and improving survival but remain constrained by challenges related to manufacturing complexity and risks of graft-versus-host disease. This review underscores the need for standardized treatment protocols as well as comparative studies to identify optimal dosing and timing to initiate treatment. Future research should focus on individualized treatment approaches and the development of novel therapeutic agents to address the unmet clinical needs of AdV management.

## 1. Introduction

Adenovirus (AdV) was first isolated from adenoidal tissues in 1953 by Wallace Rowe et al. while studying poliovirus and was termed an “adenoid degeneration agent” [1]. Shortly after, it was identified as the causative agent of an epidemic of acute respiratory illness among military service personnel [2]. Since then, this virus has been recognized as a human pathogen that can cause a myriad of clinical manifestations. While most AdV infections are mild and self-limiting, severe and fatal AdV infections began to emerge among leukemic and transplant patients in the 1970s [3] and spurred efforts to develop effective antiviral treatments for this virus.

Human AdV belongs to the family of *Adenoviridae*, which consists of viruses with non-enveloped, icosahedral virions containing linear dsDNA genomes of 25–48 kb and with the capability to infect various kinds of vertebrate hosts [4]. It is important to appreciate that human AdV is a family of viruses. As of March 2024, there are 116 “types” of human AdV identified [5], and this list will undoubtedly grow. AdV are classified into seven distinct species, from A to G, based on the guanine and cytosine content of the DNA genome. The initial 51 types were identified as “serotypes” through neutralization assays, while the remaining types were classified as “genotypes” based on bioinformatic analysis of their whole-genome sequences [6,7] (Table 1). These viruses differ not only in their serotype-specific neutralizing epitopes and/or genomic size and sequences, but they also use different receptors for cell entry and different trafficking pathways within the host cell. These properties likely play important roles in their tropism and pathogenicity [8,9]. Species A, C, D, E, and F attach to the host cells via the CAR (Coxsackie virus and adenovirus) receptor, while group B species use CD46, which is a membrane cofactor protein [8]. Different species or serotypes/genotypes are well known to affect various host populations and produce different spectrums of clinical diseases (Table 2). Notably, their responses to antiviral therapy may also vary. However, efforts to evaluate the efficacy of antivirals against specific species, serotypes, or genotypes have been limited. In clinical settings, most laboratories diagnose AdV infections by amplifying and detecting highly conserved regions of the hexon gene, but such molecular diagnostic methods lack the capability to identify the specific species, serotype, or genotype of the virus.

From multicenter studies conducted across the United States, Europe, and Japan, the incidence of AdV infection in adults undergoing hematopoietic stem cell transplantation (HSCT) ranges from 2.99% to 6%. This incidence is significantly higher in pediatric HSCT patients, ranging from 23% to 32% [11,12,13]. Similarly, in solid organ transplant (SOT) patients, the incidence of AdV infection is higher in pediatric populations compared to adults. The incidence also varies by organ type, with intestinal transplants showing a higher prevalence of infection compared to other organs [14] (Table 3).

While most AdV infections are self-limiting, they can lead to fatal outcomes, particularly in neonates and immunocompromised hosts. In immunocompromised patients, AdV infections can manifest with a wide spectrum of clinical presentations, ranging from asymptomatic viral shedding to severe disseminated disease. Various organ systems may be affected, including the central nervous system (CNS), lungs, liver, gastrointestinal (GI) tract, kidneys, and bladder. Among HSCT patients, AdV infections are associated with high mortality rates, up to 80% in some studies [15,16,17]. Mortality rates of up to 53% are reported in pediatric liver transplant patients [18].

Despite its potential to cause severe or life-threatening disease, therapeutic options for AdV infections remain limited. Clinical outcomes are highly variable based on host factors, immune status, and organs involved. Additionally, these therapies are limited by frequent and significant toxicity, which often exacerbates pre-existing end-organ dysfunction resulting from both underlying patient factors and the AdV infection itself. Current guidelines from transplant societies recommend cidofovir (CDV) as the antiviral agent of choice for AdV treatment; however, data on other antiviral drugs, such as ribavirin (RBV) and ganciclovir (GCV), as well as newer treatment options like brincidofovir (BCV) and adoptive T-cell therapy, have not been systematically evaluated [14,19,20].

This review critically explores current and emerging therapeutic options, including both antiviral agents and immune-based therapies such as adoptive T-cell treatments. In particular, we will assess the available clinical data regarding the efficacy, adverse effects, and clinical challenges associated with these anti-AdV modalities in the management of AdV infections among HSCT and SOT populations.

## 2. Cidofovir (CDV)

CDV [(S)-1-(3-hydroxy-2-phosphonylmethoxypropyl)cytosine or (S)-HPMPC] is an acyclic nucleoside phosphonate with a wide spectrum of activity against DNA viruses. De Clercq et al. first described the antiviral activity of (S)-1-(3-hydroxy-2-phosphonylmethoxypropyl)adenine [(S)-HPMPA] in 1986 [21]. In the following year, they introduced its analogue (S)-HPMPC, synthesized through substituting the adenine moiety of (S)-HPMPA with cytosine, and demonstrated its potent activity against varicella zoster virus (VZV), cytomegalovirus (CMV), and AdV [22]. CDV in its phosphorylated form is a competitive inhibitor of DNA polymerase, and once incorporated into the viral DNA, it blocks further DNA synthesis, thereby interfering with viral replication [22].

CDV was primarily studied and subsequently employed clinically for severe CMV infections in AIDS patients in the 1990s. It was FDA approved in 1996 for CMV retinitis in patients with acquired immunodeficiency syndrome (AIDS), and this remains its sole approved indication at the time of writing. CDV has been employed off-label primarily as salvage therapy or in the setting of antiviral resistance in a variety of other viral infections, including herpes simplex virus (HSV), VZV, Epstein–Barr virus (EBV), BK polyomavirus, JC polyomavirus, and so on. It has also been used as a topical preparation for HSV and human papillomavirus (HPV), as well as via intravitreal and intravesicular injection for CMV and BK virus, respectively [23,24]. Also known for its activity against poxviruses, CDV has recently been investigated for the treatment of Mpox as well as smallpox [25,26].

The major treatment-limiting toxicity of CDV is nephrotoxicity, which is dose-dependent and may manifest as increased serum creatinine, proteinuria, azotemia, and Fanconi-like syndrome with glycosuria and hypophosphatemia [27,28,29]. The nephrotoxicity of CDV is due to its extensive transport by the basolateral anion transport system of the proximal renal tubular epithelium, resulting in severe necrosis and degeneration of proximal convoluted tubule cells [30,31]. Probenecid, as a potent inhibitor of this transport system, was found in animal models and subsequently in human studies to mitigate this toxicity while not affecting CDV’s concentration in non-renal tissues [29,32,33]. Thus, a variety of toxicity-mitigating strategies, including pre-hydration, limiting dose frequency, and concomitant administration of probenecid, were employed in initial clinical trials and are still used in contemporary practice [28,29].

The in vitro activity of CDV against human AdV was first described by De Clercq et al. in 1987 in human embryonic lung cells. The minimum antiviral concentration was found to be 3.4 ug/mL, far below the minimum cytotoxic concentration, suggesting promise as a potentially viable agent against clinical human AdV infection [22]. Subsequent in vitro studies have found variable CDV IC_50_ values against human AdV but, for the most part, within clinically observable drug concentrations [34,35]. An attractive in vitro attribute of CDV is its broad activity against a wide variety of human AdV species (A through F) [36], but CDV-resistant variants could emerge by serial passaging [37].

The first use of CDV for the treatment of AdV infection was reported by Hedderwick et al. in 1998 [38]. The case involved a 40-year-old male with AIDS who developed AdV-related cholecystitis and colitis, confirmed by histopathology. The patient underwent a cholecystectomy, and his symptoms of diarrhea initially resolved with CDV administered at 5 mg/kg per week. However, the therapy was discontinued after two doses due to renal toxicity, and his diarrhea subsequently recurred. This was followed by a report of successful treatment of AdV colitis in a 17-year-old umbilical cord HSCT recipient. In this case, the patient was treated with CDV after an apparent failure of RBV therapy [39].

Subsequent to these single case reports, larger studies of AdV treatment in HSCT populations began to emerge. Legrand et al. [40] described a cohort of seven pediatric HSCT patients with AdV infection, including three with disseminated disease. Among them, CDV treatment was successful in five cases.

These early studies utilized a CDV dosing of 5 mg/kg/week intravenously (IV) for 2 weeks, then every other week (with concomitant probenecid), and associated renal toxicity of CDV was a major concern. A more “renal-protective” dosing of CDV at 1 mg/kg thrice weekly (TIW) was first reported in a prospective trial with pediatric HSCT patients [41]. Other reports using CDV at a dosage of 1 mg/kg TIW have been mostly limited to pediatric populations. Few studies [42,43] directly compared the efficacy and safety of the conventional dosing of 5 mg/kg/week versus the modified dosing of 1 mg/kg TIW. The two CDV regimens were overall well tolerated without any significant difference in nephrotoxicity, although Guerra Sanchez et al. [43] reported that the modified dosing had a higher, although non-significant, rate of viral load clearance and suggested that more frequent dosing at lower levels may be more efficacious. A recent pharmacologic review by Riggsbee et al. [44] analyzed 16 manuscripts with a total of 210 pediatric patients who received CDV for AdV. Of these, 63% of patients received the conventional dosing of 5 mg/kg/week and 37% of patients received the modified 1 mg/kg TIW dosing. Nephrotoxicity was reported in 18% of patients treated with the conventional regimen but only 4% of those on the modified regimen. A recent retrospective multicenter cohort study [45] examined the safety and efficacy of CDV in adult HSCT recipients. The study included 165 patients from nine centers who received CDV for AdV, CMV, or BK virus. Most patients (115; 69.7%) received CDV at 5 mg/kg/week, while the others received CDV at 1 mg/kg/week (18; 10.9%), 3 mg/kg/week (12; 7.3%), or 1 mg/kg TIW (11; 6.7%). Overall, 25% of these patients developed reversible nephrotoxicity, and for those who received CDV for AdV, 72% demonstrated virologic response. However, there were no comparisons made between the 5 mg/kg/week dosing and the lower doses. While it is reasonable to consider the modified dosing for adult patients with renal dysfunction, close monitoring of renal function and minimizing the use of other nephrotoxic drugs remain crucial. It is important to note that breakthrough infections with HSV and CMV have been reported with the 1 mg/kg TIW dosing [46,47] despite the fact that CDV has good activity against these herpesviruses [48].

In terms of the efficacy of CDV for AdV treatment, variable degrees of success have been reported in immunocompromised populations, ranging from 23% to 100% [41,42,49,50,51,52] for HSCT patients. Al-Heeti et al. [53] have recently reviewed studies of CDV for the treatment of AdV among SOT recipients. A majority reported successful outcomes, but this may be subject to publication bias. The discrepancies in CDV’s effectiveness are likely multifactorial. Susceptibility to CDV might vary among different species, serotypes/genotypes, or isolates. Other adjunctive therapies, such as IVIG, might be used, although the utility of IVIG for AdV treatment is debatable. The severity of AdV infections might also differ, ranging from asymptomatic viremia or viral shedding to disseminated disease. Patient characteristics also varied, such as age (pediatric versus adult), type of transplant, and degree of immune recovery. In particular, the importance of lymphocyte reconstitution has been demonstrated in several studies [51,54,55]. As such, for very high-risk patients (e.g., haploidentical HSCT), the addition of donor lymphocyte infusion (see below) to CDV treatment may confer additional benefits [56].

An important consideration regarding AdV treatment with CDV is the timing to initiate therapy. According to a survey on the incidence and management of AdV infection after allogeneic HSCT conducted among European Bone Marrow Transplant (EBMT) centers, some initiated treatment with two consecutive PCR positivity irrespective of viral load (32/74 (43%)), or based on the viral load, most frequently at >1000 copies/mL (41%), followed by 100–10,000 copies/mL (29%), >100 copies (21%), and >10,000 copies/mL (9%) [57]. In the USA, many centers use a threshold of >1000 copies/mL to initiate pre-emptive treatment in high-risk patients [13,58]. Currently, the guidelines from the European Conference for Infections in Leukemia (ECIL)-4 and the Infectious Disease Working Party (IDWP) of EBMT both recommend tapering immunosuppression as feasible and starting CDV for pre-emptive treatment in patients with AdV viral load >1000 copies/mL [19,59].

While CDV is currently considered a treatment of choice for AdV infections in immunocompromised patients, its utility is limited by renal toxicity and variable efficacy. For HSCT patients during the early post-transplant phase, early initiation of treatment to control viral replication and to allow time for immune recovery would likely be beneficial. Future studies should prioritize developing standardized treatment protocols and compare the effectiveness and side effects of dosing regimens. Additionally, tailoring treatment based on individual patient risks may further improve clinical outcomes.

## 3. Brincidofovir (BCV)

BCV (hexadecyloxypropyl-cidofovir; also known as CMX001) is a lipid conjugate form of CDV with a similar spectrum of activities against various viruses [60]. It was initially developed in response to the growing bioterrorism threat in the late 1990s and early 2000s [61]. In 2002, inspired by earlier successes in enhancing the oral bioavailability of acyclovir and ganciclovir through alkoxyalkyl esterification, Hostetler et al. applied a similar approach to cidofovir, synthesizing hexadecyloxypropyl-cidofovir—later named brincidofovir [62]. Studies in mouse models confirmed BCV’s oral bioavailability and demonstrated reduced kidney accumulation, suggesting a lower risk of nephrotoxicity [63]. When compared with CDV, BCV also had dramatically increased antiviral potency owing to a unique cellular uptake and metabolism mechanism enabling >100-fold greater intracellular active drug levels and a prolonged intracellular half-life [64,65,66].

These findings have established several key pharmacokinetic and pharmacodynamic advantages of BCV, including enhanced antiviral efficacy, oral bioavailability, and reduced nephrotoxicity. BCV was FDA approved for the treatment of smallpox in 2021 and has been procured for the US Centers for Disease Control (CDC) Strategic National Stockpile for use against smallpox and other orthopoxviruses, including Mpox, as an investigational treatment [67].

BCV’s activity against AdV was first demonstrated in 2005 by Hartline et al., who observed 5–200-fold greater antiviral potency against AdV serotypes 3, 5, 7, 8, and 31 in a human fibroblast model and then subsequently in animal models [68,69]. For clinical use, Florescu et al. [70] first reported the clinical experience of 13 immunocompromised patients who received oral BCV as emergency investigational new drug use (EIND) for treatment of AdV disease and viremia. Among these patients, 11 were allogeneic HSCT recipients, and the remaining two had severe combined immunodeficiency and small bowel transplant, respectively. In their report, 69% of patients exhibited a 99% reduction in viral load or complete resolution of viremia, with an overall survival rate of 77%. Notably, no serious adverse events were attributed to BCV, and there were no significant changes in renal function from baseline to week 8 of therapy. Subsequently, there have been a number of case reports/series as well as retrospective studies that describe the use of oral BCV for treatment of AdV infection in HSCT or SOT patients.

Among the case reports/series with 16 patients combined, 6 patients underwent SOT and 10 received allogeneic HSCT. A total of 12 were treated with CDV prior to receiving BCV. A total of 12 achieved infection resolution and survived, although 4 HSCT recipients died (Table 4). While these findings are promising, the validity of these reports may be affected by publication bias and the limited number of cases.

Multiple retrospective studies further examined the safety and efficacy of oral BCV for AdV treatment in comparison with CDV (Table 4). In a multicenter retrospective study by Hiwarkar et al. [77] with 41 pediatric and adolescent patients post-HSCT, 18 received BCV and 23 received CDV as preemptive treatment for AdV viremia. Virological response was observed in 83% of patients who received BCV compared to 9% in the CDV group. Additionally, nine out of eleven patients who did not respond to CDV had a virological response with BCV. BCV was stopped in one patient at 4 weeks due to severe abdominal cramps and diarrhea [77]. Perruccio et al. [79] reported 30 pediatric allogeneic HSCT patients with AdV reactivation (including 26 with AdV), totaling 44 episodes. CDV was used in 23 (52%) episodes as first-line treatment; BCV was used in 21 events, as first-line treatment in 7 (33%) and as rescue therapy in 14 (67%) after CDV failure. CDV treatment resulted in complete response in 35%, partial response in 4%, stable disease in 8%, and disease progression in 54% of cases. Whereas BCV treatment resulted in complete response in 48%, partial response in 9.5%, stable disease in 9.5%, and disease progression in 24%.

While these studies were limited by their retrospective nature, a phase 2 randomized placebo-controlled trial evaluated pre-emptive treatment with oral BCV for the prevention of AdV disease in pediatric and adult allogeneic HSCT recipients with asymptomatic AdV viremia [80]. Forty-eight subjects were randomized into three groups to receive either oral BCV 2 mg/kg twice weekly (BIW), BCV 4 mg/kg weekly (QW), or a placebo. After one week of therapy, undetectable AdV viremia was achieved in 67%, 29%, and 33% of patients in the BCV BIW, BCV QW, and placebo groups, respectively. Treatment failure rates were 21% for BCV BIW, 38% for BCV QW, and 33% for the placebo group. All-cause mortality was lower in the BCV BIW (14%) and BCV QW groups (31%) relative to the placebo group (39%), but not statistically significant. Diarrhea was the most common side effect reported in all three groups but led to treatment discontinuation in only one patient. Graft-versus-host disease (GVHD) of the GI tract was more common in the BCV BIW (50%) compared to the BCV QW (25%) and placebo (17%) groups. Despite its demonstrated antiviral activity against AdV, the development of oral BCV for AdV treatment has been hindered by significant GI side effects, particularly diarrhea and GVHD.

Another phase 2a study (NCT04706923) was conducted to assess the safety and efficacy of IV BCV for treatment of AdV. Preliminary results presented at the 2024 Tandem Meetings of the American Society of Transplantation and Cellular Therapy (ASTCT) and the Center for International Bone and Marrow Transplantation Research (CIBMTR) [81] demonstrated promising outcomes. From the experience of 27 immunocompromised patients, it was observed that 90% of patients achieved viral clearance in </= 4 weeks with IV BCV at a dose of 0.4 mg/kg twice weekly. Importantly, the GI and hepatic toxicities associated with oral BCV were not observed with IV BCV.

BCV has significantly enhanced oral bioavailability and reduced kidney toxicity compared to CDV. Retrospective studies suggest BCV is a viable alternative, particularly when CDV treatment fails [77,79]. Despite its efficacy, oral BCV use has been hindered by significant gastrointestinal (GI) side effects, including diarrhea and increased risk of GI graft-versus-host disease (GVHD) [80]. Unlike the oral formulation, IV BCV does not appear to cause severe GI and hepatic toxicities. Overall, these studies highlight the efficacy and improved safety profile of BCV, particularly in its IV formulation, as a promising alternative to CDV for the treatment of AdV [81].

## 4. Ribavirin (RBV)

RBV (1-β-D-ribofuranosyl-1,2,4-triazole-3- carboxamide) is a synthetic purine nucleoside analogue first synthesized in 1972 [82,83]. Unlike many other nucleoside or nucleotide analogues that exhibit antiviral activity by inhibiting viral nucleic acid replication, RBV appears to have a much more diverse mechanism of action. As outlined by Graci and Cameron [84], direct mechanisms may include inhibition of RNA capping activity, inhibition of viral polymerases, and mutagenic effects via direct incorporation of RBV into newly synthesized viral genomes. Indirect effects include the reduction in cellular GTP pools via inhibition of inosine monophosphate dehydrogenase and an immunomodulatory role that promotes a T-helper type 1 immune response.

The aerosolized form of RBV was first FDA-approved for treatment of respiratory syncytial virus in 1986, and the oral form was licensed for treatment of chronic hepatitis C in 2003. Given its very broad-spectrum antiviral activities against both DNA and RNA viruses, it has been used off-label to treat a number of human viral pathogens with various degrees of success [85]. In many of these cases, the IV form of RBV, which was not FDA-approved, was employed as investigational treatment to treat serious viral infections under an EIND application [86].

Primary toxicity concerns of RBV are anemia and teratogenicity. RBV was shown in early animal and human cell models to suppress the release of erythrocyte precursors and reduce erythrocyte survival [87]. In clinical use, this has been best described in the treatment of hepatitis C infection in the era when RBV combined with interferon was the mainstay of treatment. RBV-associated anemia appears to be dose-dependent though reversible, and may be exacerbated by pre-existing anemia and renal dysfunction (likely due to increased RBV exposure) [88]. A variety of strategies have been successfully employed to mitigate RBV-associated anemia, including dose reduction as well as administration of epoetin alfa [89]. RBV has been shown to be teratogenic in hamster and rat models, and while human data is limited with unclear association, RBV exposure should be avoided during pregnancy and in the 6 months prior to pregnancy due to prolonged residence time in erythrocytes [90,91]. This presents concern not only for the patient but for healthcare workers as well. Aerosolized ribavirin requires personal protective equipment and negative pressure rooms due to potentially toxic ambient conditions surrounding RBV concentrations during administration. Additionally, all forms of ribavirin (oral and aerosolized) should not be handled by potentially pregnant healthcare workers [92,93].

The in vitro activity of RBV against AdV was first demonstrated in cell culture models over four decades ago [94,95]. Buchdahl et al. first described the successful use of nebulized RBV for the treatment of AdV pneumonia in two children (without any immunocompromised conditions reported) in 1985 [96]. The first case of successful treatment of AdV infection with IV RBV was reported in 1991 by Cassano et al. [97]. In that case, a 9-year-old male patient developed AdV-associated acute hemorrhagic cystitis following allogeneic HSCT. His symptoms failed to respond to vigorous hydration, diuresis, and analgesic therapy, but IV RBV produced rapid resolution of symptoms and AdV viruria.

Since its initial use, numerous reports have explored RBV as a treatment for AdV infection, yielding mixed outcomes. Some studies reported successful treatment [98,99,100,101], while others presented less promising results. Table 5 summarizes the outcomes of HSCT and SOT patients treated with RBV. For instance, Hromas et al. [102] reported four sequential allogeneic HSCT recipients with AdV infection treated with IV RBV, and all four failed to clear the AdV infection. Bordigoni [103] evaluated 35 HSCT patients with AdV infection, 18 of whom were treated with IV RBV. The authors concluded that RBV was ineffective, particularly in high-risk patients prone to disseminated disease. La Rosa et al. [104] reviewed 85 adult HSCT patients with AdV infection, including 12 treated with IV RBV, and found that RBV “was not associated with an appreciable benefit.” In addition to clinical outcomes, Lankester et al. [105] prospectively measured quantitative AdV DNA load as a surrogate for treatment response. Among four pediatric allogeneic HSCT patients without immune recovery, RBV administration at the first signs of AdV dissemination did not reduce the AdV DNA load, with three patients showing increased viral loads. These collective findings cast doubt on the efficacy of RBV for treating severe AdV infections in immunocompromised patients.

More recently, the use of RBV in AdV has been reviewed by Ramfrez-Olivencia et al. [85], which included 21 isolated cases and seven case series from 1991 to 2017. These studies represented approximately 150 patients with AdV infection, though only about 60 individuals were treated with RBV (and some in combination with other antiviral agents). We conducted a focused review of cases involving HSCT and SOT recipients only, excluding those lacking sufficient data and those in which concomitant antiviral agents with potential AdV activity [such as CDV, ganciclovir (GCV), or vidarabine] were administered. As shown in Table 5, a total of 61 HSCT patients and seven SOT patients who received systemic RBV for AdV infection from 1991 to 2024 were included in our analysis. Of the HSCT cases, 22 cases (36.1%) reported successful outcomes. Among the successful cases, only four (18.2%) had disseminated disease, while the majority (59%) had AdV-associated hemorrhagic cystitis and/or nephritis. Conversely, of the 39 (63.9%) cases that resulted in treatment failure, 13 (33.3%) involved disseminated disease. The overall success rate for treating hemorrhagic cystitis and/or nephritis was 59.0%, while treatment success for disseminated disease was notably lower at only 23.5%. For the SOT patients, the overall success rate was 57.1%. Most of these patients had hemorrhagic cystitis, although one successful case involved the treatment of AdV hepatitis in a liver transplant recipient. Notably, spontaneous resolution of AdV viremia or mild end-organ disease, such as cystitis, may occur, particularly following engraftment in HSCT patients.

Despite focusing on transplant recipients, these cases still represented a highly heterogeneous population, including variations in transplant type, level of immunosuppression, severity of AdV infection, and so on. Given the discrepant results from these reports, there are several potential reasons for RBV’s inconsistent efficacy against AdV. First, as previously discussed, AdV is not a single virus but comprises seven species of many serotypes/genotypes that can cause different clinical syndromes in various host populations and may differ in their sensitivity to RBV. In 2005, Morfin et al. [36] evaluated in vitro susceptibility of AdV to RBV and CDV using reference strains. They concluded that all tested serotypes were susceptible to CDV, whereas only species C serotypes were sensitive to RBV. However, in a subsequent study involving clinical isolates [127], RBV demonstrated activity against most isolates from species A, B, and D, as well as all species C isolates. Similarly, Stock et al. found that species C was more susceptible to RBV than other species [128]. Unfortunately, most cases listed in Table 5 do not report the AdV species or serotypes, limiting the ability to establish an association between species/serotypes and clinical outcomes. At present, AdV susceptibility to RBV cannot be reliably predicted based solely on species or serotype.

Second, another possible reason for RBV failure might be related to the concentration of RBV achievable at different sites of infection. The optimal dosing regimen for RBV in treating various forms of AdV infection remains undefined, which poses a challenge in clinical practice.

Third, for treatment of any infection, the extent of the disease and the timing of treatment initiation are of crucial importance. A trend toward better response was noted among patients with a single site of infection [114]. Not surprisingly, patients with disseminated infection had the poorest outcome per our analysis.

Finally, for most infections that can cause severe or life-threatening disease, the immune status of the hosts always plays a vital role in controlling disease progression and in recovery. Some have observed better efficacy of RBV among HSCT patients with siblings as donors as compared to other donors [116]. Patients with concomitant acute GVHD or a long delay between infection and treatment were found to be at greater risk of treatment failure [103].

Therefore, firm conclusions about RBV’s efficacy for AdV treatment remain uncertain in the absence of prospective clinical trials. Further research is needed to clarify its role in the management of AdV infections.

## 5. Ganciclovir (GCV)

The activities of GCV [9-(1,3-dihydroxy-2-propoxymethyl guanine)] against different serotypes of AdV were demonstrated in 1988 using an in vitro plaque reduction assay [129]. The 50% effective dose (ED50) for AdV ranged from 4.5 to 33 μM, making it approximately 6 to 43 times less potent for AdV than for CMV. However, the drug concentrations achievable in patients receiving GCV for CMV treatment suggested potential efficacy against AdV infection. Subsequent in vitro and animal models also demonstrated GCV (or valganciclovir, VGCV)’s activity against AdV [130,131,132], but reports of its clinical use against AdV remain scant. In a systematic review by Gu et al. [133] that included 228 cases of AdV disease from 2000 to 2019, only 18 cases (7.9%) were treated with GCV. This was notably lower compared to other antiviral agents, with 36.0% of cases treated with CDV, 6.6% with RBV, and 5.3% with BCV.

The first case of successful AdV treatment with GCV was reported in 1992 with a renal transplant recipient who suffered from AdV-associated hemorrhagic cystitis [134]. Two other successful cases were then reported in 1997, including an HSCT patient with AdV-associated hemorrhagic cystitis and a cardiac transplant patient with severe AdV pneumonia [135,136]. A total of 21 cases using GCV for AdV treatment in HSCT or SOT recipients are found in the literature (Table 6). A majority (90%) of these cases reported successful outcomes. Among 16 HSCT or kidney transplant recipients presenting with nephritis or cystitis (including four with viremia and one with pneumonia concomitantly), treatment was successful in all but one case. However, for the four HCT patients with disseminated disease, only half survived despite treatment.

The apparent success of GCV for the treatment of hemorrhagic cystitis and/or nephritis warrants further discussion. GCV is highly concentrated in the kidney tissues, and ~90% of the daily dose of GCV is excreted unchanged in the urine [147]. The GCV concentration in urine is also substantially higher than that in serum. For instance, in the case report by Nakazawa et al. [138], a 9-year-old girl who developed AdV-associated hemorrhagic cystitis after HSCT was successfully treated with GCV [138]. The authors noted that the peak serum concentration of GCV was 42.2 μM, but the concentration of GCV in urine exceeded 300 μM for nearly half a day after the infusion. Thus, despite the high EC_50_ of GCV for AdV, the very high drug concentration in urine might contribute to successful treatment outcome. Another piece of evidence suggesting potential clinical utility of GCV against AdV is based on the observation that HSCT patients receiving GCV for CMV prophylaxis had a lower risk for the development of AdV infection or for progressive AdV disease [148,149].

The major dose-limiting toxicity of GCV is hematologic toxicity, primarily neutropenia, occurring in more than 35% of patients receiving treatment-dose GCV therapy [150]. GCV-associated neutropenia is reversible in most cases and may be ameliorated with the administration of granulocyte colony-stimulating factor, permitting prolonged treatment [151].

Overall, the most robust data regarding the use of GCV in the treatment of AdV is primarily limited to cases of nephritis or cystitis. However, as previously discussed, hemorrhagic cystitis may resolve spontaneously without the need for specific antiviral treatment.

## 6. Other Anti-AdV Agents

Historically, vidarabine (AraA; 9-β-D-arabinofuranosyladenine) was also used clinically to treat AdV infection. It was the first FDA-approved nucleoside analogue to be administered systemically and was licensed in the United States in 1977 for the treatment of life-threatening HSV and VZV infections. However, given more favorable toxicity profiles of newer anti-herpes agents such as acyclovir, IV vidarabine was discontinued in the US in 2001, although it remains available as an ophthalmic ointment indicated for acute keratoconjunctivitis and recurrent epithelial keratitis secondary to HSV. Experience with this drug for AdV treatment is limited to only a handful of reports [11,103,122]. Most cases of success were for treatment of AdV-associated hemorrhagic cystitis [152,153,154] and pharmacokinetic data did support its possible efficacy in this setting [155].

Currently, there are no antiviral drugs approved for the treatment of AdV diseases despite their significant morbidity and mortality in vulnerable patient populations. However, there are continued efforts to identify effective anti-AdV therapies [156,157]. While various agents have demonstrated in vitro activities, none of these have been utilized for AdV treatment in clinical settings to date [158,159,160].

## 7. Adoptive T-Cell Therapy

As discussed above, conventional antiviral agents used for the treatment of AdV are limited by both unreliable efficacy and/or toxicity. The recognition that antiviral drugs were ineffective, especially for severe or disseminated AdV infections, highlights the need for immune-based approaches. Early research identified the critical role of T-cells in controlling AdV infections, particularly in immunocompromised individuals [51]. For instance, in a prospective study of renal transplant patients with AdV infection, an absolute lymphocyte count of <300 cells/µL was identified as a predictor of poor outcome, while an increase in virus-specific CD4+ and CD8+ T-cell counts was associated with successful viral clearance [161].

Adoptive T-cell therapy involves the transfer of virus-specific T-cells (VSTs) from a donor to a patient to enhance the immune response against infections. Donor lymphocyte infusions were initially used in the 1990s to treat viral infections in immunocompromised individuals [162,163]. However, these infusions often led to GVHD as a complication. Over the subsequent years, this field has evolved significantly with advancements in techniques for isolating and expanding T-cells, resulting in more effective and targeted therapies for AdV and other viral infections. A detailed description of the manufacturing of VSTs is below the scope of this review and has been well described in the literature [164].

The first step in the process of manufacturing VSTs involves selecting a donor. For HSCT patients, the cells can be obtained from the stem cell transplant donor or 3rd-party healthy donors. The safety and efficacy of donor-derived VSTs for AdV treatment have been evaluated in multiple studies (Table 7).

Virus-specific T-cell therapy has also been used to treat refractory infections in SOT patients [179]. In 2006, Leen et al. demonstrated that multivirus-specific T-cells targeting CMV, EBV, and AdV (derived from a single culture and expanded in patients) led to reductions in viral titers and resolution of associated symptoms [180]. Additionally, the safety of EBV- and AdV-specific T-cells was shown in 20 pediatric patients who had undergone haploidentical or matched unrelated donor transplants [165]. None of these patients developed EBV proliferative disease, and two had resolution of AdV infection without any reported GVHD. Similarly, a clinical trial in Germany evaluated the safety and efficacy of hexon-specific T-cell therapy for AdV infections. Among 14 patients, VSTs induced in vivo antiviral immunity lasting up to six months, with viral control leading to complete clearance of viremia in 86% of patients with antigen-specific T-cell responses. Six-month survival was markedly higher in responders compared to non-responders, who all died shortly after adoptive T-cell therapy. GVHD grade 1 occurred in two patients within 2 weeks and grade 2–3 in four patients at approximately seven weeks after VST administration. Although the late onset of GVHD suggested other possible causes, the role of VSTs could not be definitively excluded [168]. Ip et al. conducted a phase 1/2 open-label trial to evaluate the safety and efficacy of AdV-specific T-cells in high-risk pediatric patients. All eight patients cleared viremia between days 56 and 127. AdV-specific T-cells were detectable until day 90 in all patients via ELISpot assay. However, one patient developed GVHD requiring steroid treatment, which led to AdV reactivation, respiratory failure, and death [170]. The efficacy of VSTs in preventing viral infections in immunocompromised patients has also been assessed. In a clinical trial by Rubinstein et al. [172], 23 patients received VSTs targeting CMV, AdV, BKV, and EBV on day 21 post-transplantation to assess their efficacy in preventing viral infections. Of these, 18 did not develop infections, 2 patients experienced EBV viremia, 1 developed symptomatic BK viruria, 1 developed CMV viremia, and 2 developed clinically significant GVHD. While this study demonstrated effectiveness, it was limited by its small sample size and lack of a control arm [172].

Overall, these studies have demonstrated the safety and efficacy of donor-derived virus-specific T-cells, with significant reductions in viral loads, resolution of symptoms, and minimal GVHD in some trials. Nevertheless, challenges such as late-onset GVHD, small sample sizes, and the absence of control arms in many trials are some of the limitations.

Despite the successful use of adoptive T-cell therapy in many of these cases, the generation of VSTs for each individual patient from the stem cell donor is a time-consuming process and requires the donor to be seropositive for AdV. It may also increase the risk of GVHD if the cells are directly isolated from donor leukocytes using methods like antigen capture or using viral peptides. Furthermore, this approach may not be practical for urgent or widespread use. These limitations led to exploration of the use of 3rd-party healthy donors to generate VSTs [175,181]. Third-party VSTs can be partially HLA-matched and may be used for multiple recipients. However, donor-derived VSTs may have longer persistence due to a higher degree of human leukocyte antigen (HLA) matching compared to third-party VSTs.

In 2013, Leen et al. generated a third-party bank of VSTs and conducted a multicenter clinical trial to assess their efficacy [173]. The third-party VSTs were matched at one HLA allele. Among 50 patients, 18 had AdV infections. Of these, 77% experienced partial or complete responses within six weeks post-infusion. Across the entire cohort, two patients developed de novo GVHD, but no other toxicities were reported.

To identify potential donors for T-cell therapy, Li Pira et al. [182] measured donor T-cell responses to different viral antigens using a cell-ELISA assay. They demonstrated a strong correlation between the frequency of specific T-cells and the cell-ELISA results, which is useful for selecting the best donors. Based on their findings, they advocate for the creation of registries for third-party donors who are HLA-typed and fully characterized for pathogen-specific T-cell immunity. This approach would expand the use of third-party donors and increase the likelihood of better HLA matching.

In 2021, the FDA granted orphan drug designation to Posoleucel, a multivirus-specific T-cell therapy derived from partially HLA-matched third-party donors. Posoleucel was designed to prevent or treat multiple viruses, including AdV, BKV, CMV, EBV, HHV-6, and JC virus. In an open-label, single-arm phase 2 study, 59 cell lines were administered to 58 patients [175]. Overall, 55 out of 59 patients showed partial or complete responses at six weeks post-infusion. Among the 12 patients with AdV infections, 19 infusions were administered, with a response rate of 83% (10 of 12) observed by week 6. Thirteen of 58 patients (22%) developed GVHD. This is one of the largest reported trials demonstrating the safety and efficacy of third-party VSTs for the treatment of viral infections.

The promising results from the above trial prompted another phase 2 open-label single-arm study [183] (NCT04693637) to evaluate the safety and efficacy of posoleucel in preventing six viral infections. Of the 26 patients enrolled, 3 (12%) had clinically significant infection after 14 weeks, and 5 (19%) patients had grade 2–4 GVHD. T-cell responses persisted until week 14 as measured by deep sequencing. Six patients died due to disease relapse or progression. This study was limited by its small sample size, the absence of a comparison group, and the fact that the posoleucel infusion was administered relatively late, with a median of 42 days after HSCT [183]. In the phase 3 portion of this trial (NCT05305040), 377 patients were enrolled; however, the study was terminated early in December 2023 due to futility. No safety concerns were identified. Another randomized, placebo-controlled trial investigated the use of posoleucel for treating AdV infections in pediatric and adult patients following HSCT. However, this trial was terminated early due to its failure to meet the predefined endpoint (NCT05179057).

There are very few studies that have compared the efficacy of donor-derived vs. third-party VSTs (Table 7). The largest study reported involved 145 children at Cincinnati Children’s Hospital who received VSTs to treat AdV, BK virus, CMV, and/or EBV. This retrospective study [178] compared the clinical efficacy and safety outcomes of donor-derived VSTs and third-party VSTs. No statistically significant differences were observed in clinical response rates between the donor-derived and third-party cohorts (65.6% versus 62.7%), incidence of GVHD, or overall survival at 30, 100 days, and one year post-transplantation.

In their review, O’Reilly et al. [181] describe the experiences of multiple centers in creating banks of varying sizes containing EBV-, AdV-, and CMV-specific cell lines. The advantage of these banks is that the cells are readily available for use. They are characterized by their HLA types, which allows for the selection of appropriate HLA-restricted T-cells for patient treatment [181]. However, the feasibility of maintaining and implementing these banks may pose challenges for many centers.

VSTs have shown significant efficacy in treating AdV and other viral infections in immunocompromised patients, with viral clearance rates of 77–94% reported in clinical trials. While VST therapy is generally well-tolerated, for HSCT recipients, GVHD remains a notable complication. Currently, the widespread adoption of VST therapy is hindered by the lack of randomized controlled trials to comprehensively assess its safety and efficacy, as well as the complexity of its manufacturing process. Future studies with larger cohorts, robust designs, and standardized endpoints are essential to establish its clinical utility. Additionally, CRISPR technology is being leveraged to enhance the precision and efficacy of VSTs by improving specificity, minimizing off-target effects, and engineering resistance to viral immune evasion mechanisms [184].

## 8. Conclusions

AdV infections pose a major clinical challenge, especially in immunocompromised populations, resulting in significant morbidity and mortality. Currently, CDV remains the primary antiviral agent of choice despite its modest efficacy and dose-limiting nephrotoxicity. BCV offers a promising alternative with a more favorable safety profile, particularly in its IV formulation. RBV and GCV have demonstrated mixed efficacy, exhibiting variable responses across different clinical diseases and possibly depending on specific AdV serotypes/genotypes. Adoptive T-cell therapy has emerged as a transformative approach for managing severe and refractory AdV infections, providing enhanced viral clearance and improved clinical outcomes. However, challenges such as the complexity of T-cell manufacturing and the risk for GVHD remain barriers to widespread adoption. Future research should prioritize the development of standardized treatment protocols and the conduct of robust comparative studies to determine optimal dosing regimens and the timing for initiating treatment. Additionally, therapies must be tailored to individual patients. There is a pressing need for ongoing research into safer and more effective therapeutic options, including novel antiviral agents and immune-based therapies, to improve patient outcomes in the management of severe AdV infections.

## Figures and Tables

**Table 1 viruses-17-00599-t001:** Classification of human adenovirus (hAdV) serotypes and genotypes.

Species		Human Serotypes and Genotypes
hAdV-A	*Mastadenovirus adami*	12, 18, 31, 61
hAdV-B	*Mastadenovirus blackbeardi*	**Subspecies B1**: 3, 7, 16, 21, 50, 64, 66, 68, 76, 114 B1/B2 recombinants: 77, 78 **Subspecies B2**: 11, 14, 34, 35, 55, 79, 106
hAdV-C	*Mastadenovirus caesari*	1, 2, 5, 6, 57, 89, 104, 108
hAdV-D	*Mastadenovirus dominans*	8, 9, 10, 13, 15, 17, 19, 20, 22–30, 32, 33, 36–39, 42, 49, 51, 53, 54, 56–60, 62, 63, 65, 67, 69, 70, 71–73, 74, 75, 80–88, 90–103, 105, 107, 109, 110, 111, 112, 113, 115, 116
hAdV-E	*Mastadenovirus exoticum*	4
hAdV-F	*Mastadenovirus faecale*	40, 41
hAdV-G	*Mastadenovirus russelli*	52

(Adapted from Kajon, A. E., 2024 [7], and personal communication.) Classification of hAdV serotypes and genotypes described to the present. -- Types originally described as “serotypes” based on their distinct antigenic reactivities in neutralization assays. Currently also designated as genotypes 1–51. -- and --Intertypic recombinant genotypes. -- Intertypic recombinant genotypes with novel hexon genes. 52: Genotype of probable simian origin.

**Table 2 viruses-17-00599-t002:** Clinical diseases caused by adenovirus infection.

Clinical Disease	Populations at Risk	Causal Adenovirus types
Pharyngitis	Infants, children	1–7
Pharyngoconjunctival fever	Children	3, 7
Pertussis-like syndrome	Children	5
Pneumonia	Infants, children Military recruits	1–3, 21, 56 4, 7, 14
Acute respiratory disease	Military recruits	3, 4, 7, 14, 21, 55
Conjunctivitis	Children	1–4, 7
Epidemic keratoconjunctivitis	Adults, children	8, 11, 19, 37, 53, 54
Gastroenteritis	Infants Children	31, 40, 41 2, 3, 5
Intussusception	Children	1, 2, 4, 5
Hemorrhagic cystitis	Children HSCT, renal transplant recipients	7, 11, 21 34, 35
Meningoencephalitis	Children, immunocompromised hosts	2, 6, 7, 12, 32
Hepatitis	Pediatric liver transplant recipients	1–3, 5, 7
Nephritis	Renal transplant recipients	11, 34, 35
Myocarditis	Children	7, 21
Urethritis	Adults	2, 19, 37
Disseminated disease	Neonates, immunocompromised hosts	1, 2, 5, 11, 31, 34, 35, 40

Adapted from Mandell, Douglas, and Bennett’s *Principles and Practice of Infectious Diseases* (2020; 9th edition), Chapter 142 on adenoviruses [10]. Abbreviations: HSCT, hematopoietic stem cell transplant.

**Table 3 viruses-17-00599-t003:** Incidence of adenovirus infection by organ transplanted.

Allograft Type	Reported Adenovirus Incidence
**Pediatric Transplantation**	
Liver	3.5–38%
Heart, heart–lung, lung	7–50%
Kidney	11%
Intestinal, multivisceral	4.3–57.1%
**Adult Transplantation**	
Liver	5.8%
Heart, heart–lung, lung	6–22.5%
Kidney	4.1–6.5%
Intestinal, multivisceral	NA

Adapted from Florescu et al., 2019 [14].

**Table 4 viruses-17-00599-t004:** Transplant patients with adenovirus infection treated with brincidofovir.

**First Author [Reference]**	**Year of Publication**	**Number of Patients**	**Age/Sex**	**Underlying Condition**	**Onset of AdV from Tx**	**AdV Infection**	**Species/Serotype**	**Other Treatment**	**Adjunctive Measure**	**Outcome**	**Comments**
**HCT Cases:**											
Paolino [71]	2011	1	12 Y/F	Allo-HCT for aplastic anemia	D + 89	GI, liver, Lung	NA	CDV prior	IVIG, Failed CDV prior, Reduction in IS	Recovery	
Voigt [72]	2016	1	5 Y/F	Allo-HCT X 2 (MUD) for MDS	D + 237	Viremia	C	CDV prior	Failed CDV prior	Recovery	Also had resistant HSV-1 infection that resolved
Ramsay [73]	2017	3	59/F	MUD Allo for AML	D + 22	Lung, GI	NA	CDV prior	Reduction in IS	Recovery	
			57/F	MRD Allo for MM	D + 51	Lung, GI				Died, had viremia at time of death	
			34/M	Mismatched Allo for ALL	D + 21	GI, hepatic, urine			Reduction in IS	Recovery	
Meena [74]	2019	5	2.2 Y/M	MUD for MDS	D + 17	GI	NA	CDV prior		Recovery	Concomitant CMV and PIV infection
			11 Y/F	MUD for ALL	D + 18	GI, Lung		CDV prior		Died of sepsis	Concomitant CMV and rhinovirus infection
			10 Y/F	MRD for osteopetrosis	D + 38	GI, Lung		CDV prior		Recovery	Concomitant PIV, rhinovirus and RSV
			15 Y/F	Cord Blood transplant for AML	D + 303	GI		CDV prior	IVIG	Died of sepsis	Concomitant rhinovirus, sapovirus and EBV
			2.9 Y/M	MUD for CDA type 2	/	GI				Died of EBV pneumonitis	
**SOT Cases:**											
Sulejmani [75]	2018	2	44 F	Intestinal Tx	D + 30	GI	NA	Ribavirin and CDV prior	Reduction in IS	Recovery, 2 episodes of rejection	
			28 M	Intestinal Tx	6 years from Tx	GI		CDV prior	Reduction in IS	Recovery	
Londeree [76]	2020	4	17 M	Kidney Tx	D + 12	Bladder	NA	NA	Reduction in IS	Recovery	
			19 M	Kidney Tx	D + 912	Lung		CDV prior	Reduction in IS	Recovery	
			13 M	Liver-kidney Tx	D + 487	Bladder, kidney		CDV prior	IVIG, Reduction in IS	Recovery	
			9 mo F	Liver Tx	D + 33	Viremia, Liver		CDV prior		Recovery	
**Clinical** **Studies** **(>10 patients)**											
**Author [Reference]**	**Year**	**Type of study**	**No. of** **Patients**	**Median age (range)**	**Onset of AdV from Tx**	**Underlying condition**	**Treatment agents**	**Virologic response**	**Outcome**	**Adverse events/side effects**	**Comments**
Florescu [70]	2012	Retrospective Multi-center	13	6 (0.92–66)	D + 75 (15–720)	11 Allo HSCT, 1 SCID, 1 Intestinal Tx	All patients = CDV + BCV	A total of 9/13 patients (69.2%) achieved a VR * at week 8	The 8-week survival rate was 76.9%	None attributed to BCV	Compared with non-responders, complete responders had longer survival (median, 196 days versus 54.5 days; *p* = 0.04)
Ramsay [73]	2017	Retrospective Single-center	10	40 (17–66)	D + 65 (20–1140)	HSCT	2 = BCV 5 = CDV 1 = BCV + CDV	Complete VR in 2/3 who received BCV, 4/6 in CDV	Two patients survived and one died in BCV group. Four patients survived and two died in CDV group.	NR	
Hiwarkar [77]	2017	Retrospective Multi-center	41	5 (2 months to 18 years)		HSCT	18 = BCV 23 = CDV	Complete VR in 15/18 (83%) who received BCV and 2/23 9% in CDV	Thirty-nine patients survived and two patients who received CDV died from AdV infection	Severe abdominal cramps and diarrhea in 1 patient who received BCV	BCV led to major responses in 9 of 11 CDV-unresponsive patients
Thomas [78]	2021	Retrospective Single-center	93	4.07 (2.12, 10.5)	D + 60 (IQR 25–75%: 16–123 days)	HSCT	5 = BCV 13 = CDV 6= Both BCV + CDV 66= no Tx	A total of 4/5 in BCV and 5/6 in BCV + CDV had resolution of disease	A total of 47/93 (51%) patients died in the total cohort, no data reported separately about BCV or CDV	No statistically significant difference in the hazard of rise in creatinine, elevated LFTs or diarrhea between groups	
Perrucio [79]	2021	Retrospective Multi-center	30 patients with 44 episodes of AdV infection	10 (9 months–19 years)	D + 90 (20 days–14 months)	HSCT	23 = CDV 21 = BCV	In BCV group: CR 48% PR ** 9.5% Stable 9.5% Progressive 24% In CDV group: CR 36% PR 4% Stable 8% Progressive 54%	Overall survival was 30%, 13% mortality due to AdV	1 patient with GI toxicity related to BCV (5%)	The response rate was higher with BCV compared to CDV (67% vs. 47%, *p* = 0.05)
**Clinical trial**											
**Author [Reference]**	**Year**	**Type of study**	**No. of** **Patients**	**Median age (range)**	**Onset of AdV from Tx**	**Underlying condition**	**Treatment agents**	**Virologic** **response**	**Outcome**	**Adverse events/side effects**	**Comments**
Grimley [80]	2017	Randomized placebo controlled phase 2 trial for prevention of adenovirus disease in patients with adenoviremia	48	8 (0–55) in BCV 2 mg/kg BIW group 9 (2–70) in BCV 4 mg/kg QW group 11 (1–53) in placebo group	NA	HSCT	14 = BCV BIW 16 = BCV QW 18 = Placebo	67% in BCV BIW group 29% in BCV QW group 33% in placebo group	**Treatment failure:** 21% in BCV BIW group 38% in BCV QW group 33% in placebo group **All-cause mortality:** 14% in BCV BIW group 31% in BCV QW group 39% in placebo group	**Diarrhea** 57% in BCV BIW group 38% in BCV QW group 28% in placebo group **GI GVHD** 50% in BCV BIW group 25% in BCV QW group 17% in placebo group	No myelotoxicity or nephrotoxicity

* Virologic response (VR) was defined as achievement of ≥99% decrease in plasma viral load from baseline or undetectable viral load by the end of treatment or follow-up period. ** Partial response (PR) was defined as a >1 <2-log reduction in viral load by the end of treatment. **Abbreviations:** AdV, adenovirus; Allo-HCT, allogeneic hematopoietic cell transplant; ALL, acute lymphocytic leukemia; AML, acute myeloid leukemia; BCV, brincidofovir; BIW; twice weekly; CDV, cidofovir; CMV; cytomegalovirus; CR, complete remission; EBV, Epstein–Barr virus; GI, gastrointestinal; GVHD, graft-versus-host disease; HSV, herpes simplex virus; HSCT, hematopoietic stem cell transplant; IQR, interquartile range; IS, immunosuppression; IV, intravenous; IVIG, intravenous immunoglobulin; LFT, liver function tests; MDS, myelodysplastic syndrome; MM, multiple myeloma; mo, months; MRD, matched related donor; MUD, matched unrelated donor; NA, not available; PIV, parainfluenza virus; PO, per os; PR, partial response; pt, patient; QW, once weekly; RSV, respiratory syncytial virus; SCID, severe combined immunodeficiency disorder; SOT, solid organ transplant; Tx, transplant; VR, virological response.

**Table 5 viruses-17-00599-t005:** Transplant patients with adenovirus infection treated with ribavirin.

First Author [Reference]	Year of Publication	Number of Patients	Age/Sex	Underlying Condition	Onset of AdV from Tx	AdV Infection *	Species/Serotype	RBV Route/Dose/Duration (If Available)	Adjunctive Measure	Outcome **	Comments
**HCT Cases:**											
Cassano [97]	1991	1	9 Y/M	Allo-HCT (MRD) for AML	D + 36	Hemorrhagic cystitis	NA	IV; 33 mg/kg/d × 1 d, followed by 16.6 mg/kg/d × 8 d; daily dose divided into three doses given q8 h		Recovery	
Liles [106]	1993	1	25 Y/M	Allo-HCT (MMRD) for T-cell ALL	D + 75	Nephritis	B11	IV; 35 mg/kg/d × 1 d, followed by 25 mg/kg/d × 9 d; daily dose divided into three doses given q8 h	IVIG	Recovery	
Murphy [107]	1993	1	8 Y/M	Allo-HCT (Haploid) for acute nonlymphocytic leukemia	D + 103	Hemorrhagic cystitis	NA	IV; 35 mg/kg/d × 1 d, followed by 25 mg/kg/d × 8 d; daily dose divided into three doses given q8 h	/	Recovery	
Hromas [102]	1994	4	7 Y–45 Y	Allo-HCT (MUD) for NHL (1), MDS (1), ALL (2)	D + 20 to D + 147	GI; GU; GI + GU; disseminated	B11 (3 cases); A12 (1 case)	“Based on that recommended by Cassano [90]”.—see above	IVIG (all)	Failure (all four pts), two died from AdV	
Jurado [108]	1995	1	27 Y/M	Allo-HCT (MRD) for aplastic anemia	D + 9	Hemorrhagic cystitis	B11	IV; 35 mg/kg/d × 1 d, followed by 25 mg/kg/d × 8 d; daily dose divided into three doses given q8 h	/	Recovery	
Kapelushnik [109]	1995	1	3 Y/M	Allo-HCT (MUD) for Wiscott-Aldrich syndrome	D + 45	Gastroenteritis	NA	IV; 30 mg/kg/d × 10 d; daily dose divided into three doses	IVIG	Recovery	
Wulffraat [110]	1995	1	8 M/M	Allo-HCT (Haploid) for SCID	D + 6	PNA, GI	NA	IV; loading dose 30 mg/kg; maintenance 15 mg/kg q6 h × 14 days	/	Recovery	
Mann [111]	1998	1	37 Y/F	Allo-HCT (MMRD) for AML	D + 33	Disseminated	NA	IV; 33 mg/kg q6 h × 5 d, followed by 16 mg/kg q6 h × 4 d, then 8 mg/kg q8 h × 1 d	/	Failure, died from AdV	
Chakrabarti [112]	1999	1	44 Y/M	Allo-HCT (MUD) for CML	D + 210	Hepatitis	NA	IV; loading dose of 35 mg/kg, followed by 25 mg/kg q8 h	/	Failure, died from AdV	
Hale [113]	1999	2	Pediatric	HCT	NA	PNA; hemorrhagic cystitis	NA	IV	/	Failure, both died from AdV	
Howard [114]	1999	6	NA	Allo-HCT (partially matched)	NA	Hemorrhagic cystitis, nephritis	NA	IV	Some also received IVIG, but no detailed information available.	Failure	Thirteen pts were treated with RBV in this cohort, but results are available only for six pts.
				Allo-HCT (MUD)		GI, GU				Failure, died from AdV	
				HCT		URT, GI				Recovery	
				Allo-HCT (MRD)		Sputum, blood				Recovery	
			Adult	Auto-HCT		Hemorrhagic cystitis				Recovery	
			Pediatric	Allo-HCT (partially matched)		Hemorrhagic cystitis				Recovery	
Lakhani [115]	1999	1	26 Y/F	Allo-HCT (MUD) for CML	D + 38	Hemorrhagic cystitis	NA	PO; 1 g bid × 8 d	/	Recovery	AdV viruria resolved with RBV, but pt received E-aminocaproic acid to treat intractable hematuria.
Miyamura [116]	2000	8	11 Y–34 Y	HCT	NA	All with hemorrhagic cystitis except one also had PNA	NA	IV; For adults, 16 mg/kg q6 h × 4 d, followed by 8 mg/kg q8 h × 3 d. For children, 15 mg/kg/d × 10 d	/	Recovery (3); Failure (5), two died from ADV.	A total of nine pts in this cohort but one pt had early death from progression of underlying disease.
Bordigoni [103] ***	2001	13	Pediatric and adult	Allo-HCT [MUD (10); MRD (2); MMRD (1)]	D + 0 to D + 184	Definite (3); Probable (5); Asymptomatic (5)	NA	IV; loading dosage of 35 mg/kg followed by 25 mg/kg q8 h × 10 d.	/	Recovery (3), among one probable and two asymptomatic cases; Failure (10)	
La Rosa [104]	2001	12	18 Y–59 Y	NA	NA	PNA (2); hemorrhagic cystitis (1), enteritis (1), disseminated (8)	NA	IV	/	Recovery (2). But both with disseminated disease	
Ikegame [117]	2001	1	50 Y/M	Allo-HCT (MMRD) for CML	D + 13	Disseminated	NA	PO; 1200 mg/d in divided doses × 4 d	IV vidarabine prior to RBV	Failure, died from AdV	
Gavin [118]	2002	1	18 Y/M	Allo-HCT (MUD) for AML	D + 15	Disseminated	B34	IV; 33 mg/kg on day 1, followed by 16 mg/kg q6 h × 3 d, then 8 mg/kg q8 h × 3 d	/	Failure, died from AdV	
Aebi [119] *	2003	1	41 Y/M	Allo-HCT (MUD) for ALL	D + 50	Hemorrhagic cystitis	B11	PO; 1st course: 16 mg/kg q6 h × 4 d, followed by 8 mg/kgq6 h × 3 d. 2nd course: 20 mg/kg q6 h × 5 d, followed by 10 mg/kg q6 h × 4 d.	/	Recovery	The pt had clinical improvement and reduction in AdV titer in urine after the 1st RBV course. When AdV titer increased and symptoms recurred 2–4 weeks later, the pt received a 2nd course of RBV, and responded well with resolution of AdV infection.
Omar [120]	2010	2	31 Y/F	Allo-HCT	D + 150	GI; viremia	C5	IV	/	Failure (2), both died from AdV	
			48 Y/M		D + 28	Disseminated	B35				
Sahu [121]	2016	1	36 Y/M	Allo-HCT for AML	D + 380	Hemorrhagic cystitis	NA	PO; 20 mg/kg in two divided doses × 4 weeks	Reduction in immunosuppression for GVHD	Recovery	
Takada [122]	2024	1	31 Y/M	HCT (cord-blood) for MPAL	D + 15	Hemorrhagic cystitis	NA	PO; 1200 mg/d × 44 d		Recovery	Received 5 days of GCV + vidarabine without effect before switching to RBV
**SOT Cases:**											
Arav-Boger	2000	1	13 M/F	Liever Tx ×2, 5 days apart	6 days from 2nd Tx	Hepatitis	C5	IV; Loading dose of 33 mg/kg, then 16 mg/kg q6 h × 4 d, followed by 8 mg/kg q8 h × 6 d	Reduction in immunosuppression	Recovery	Also received GCV and CMV IgG prior to RBV
Gavin [118]	2002	2	5 Y/F	Heart Tx	2 months post Tx	Cystitis, neprhitis	NA	IV; 25 mg/kg in three divided doses on day 1, then 15 mg/kg/d divided q8 h × 9 d		Failure (2), both died from AdV	
			2 M/M		5 weeks post Tx	PNA					
Emovon [123]	2003	1	46 Y/F	Kidney and Pancreas Tx	22 months post tx	Hemorrhagic cystitis	NA	IV	Reduction in immunosuppression; IVIG	Recovery	
Hofland [124]	2004	1	60 Y/M	Liver Tx (1995), kidney Tx (2002)	2 moths post kidney tx	Nephritis, hemorrhagic cystitis	NA	NA; 400 mg bid × 3 weeks	High-dose prednisone	Recovery	
Komiya [125]	2009	1	63 Y/F	Kidney Tx	D + 7	Nephritis, hemorrhagic cystitis	B11	NA	IVIG; reduction in immunosuppression	Failure	
Park [126]	2015	1	32 Y/F	Kidney Tx	10 months post Tx	Nephritis, hemorrhagic cystitis, viremia	NA	NA; 400 mg bid × 3 weeks	IVIG; reduction in immunosuppression	Recovery	

* Disseminated infection is defined by involvement of two or more end-organs. Involvement of a single end-organ with viremia is not classified as disseminated disease. ** Failure is defined as lack of clinical and/or virologic response to treatment. For those patients with death reported, cause of death was attributed to AdV infection only for some. *** Pts that received other antiviral agent(s) with potential activities against AdV or donor leukocyte infusion in addition to ribavirin were excluded. **Abbreviations:** AdV, adenovirus; ALL; acute lymphocytic leukemia; Allo-HCT, allogeneic hematopoietic cell transplant; AML, acute myeloid leukemia; Auto-HCT, autologous hematopoietic cell transplant; CML, chronic myeloid leukemia; CMV, cytomegalovirus; GCV, ganciclovir; GI, gastrointestinal; GU, genitourinary; HCT, hematopoietic cell transplant; IgG, immunoglobulin; IV, IV; IVIG, IV immunoglobulin; MDS, myelodysplastic syndrome; MMRD, mismatched, related donor; MPAL, mixed-phenotype acute leukemia; MRD, matched related donor; MUD, matched unrelated donor; NA, not available; NHL, non-Hodgkin lymphoma; PNA, pneumonia; PO, per os; pt, patient; RBV, ribavirin; SCID, severe combined immunodeficiency; SOT, solid organ transplant; Tx, transplant; URT, upper respiratory tract.

**Table 6 viruses-17-00599-t006:** Transplant patients with adenovirus infection treated with (val)ganciclovir.

First Author [Reference]	Year of Publication	Number of Patients	Age/Sex	Underlying Condition	Onset of AdV from Tx	AdV Infection *	Species/Serotype	(V)GCV Route (If Available) **	Adjunctive Measure	Outcome ***	Comments
**HCT Cases:**											
Chen [135]	1997	1	47 Y/M	Allo-HCT (MRD) for AML	D + 52	Hemorrhagic cystitis	NA	IV GCV	/	Recovery	
Suzuki [137]	2008	1	35 Y/F	Allo-HCT (mismatched) for MDS	D + 24	Disseminated	B3 and B34	GCV		Recovery	
Nakazawa [138]	2009	1	8 Y/F	Allo-HCT (haploid) for AML	D + 24	Hemorrhagic cystitis	B11	IV GCV	/	Recovery	
Mochizuki [139]	2014	2	50 Y/M; 41 Y/F	Allo-HCT (MUD) for MM; Allo-HCT for AML	D + 20; D + 427	Disseminated (2)	NA	GCV		Failure, both died from AdV	
Yasuda [140]	2019	1	66 Y/F	Auto-HCT for MM	D + 46	Disseminated	NA	GCV	Discontinued of pomalidomide and dexamethasone	Recovery	GCV was started for CMV viremia, not intended for AdV
Takada [122]	2024	1	31 Y/M	HCT (cord-blood) for MPAL	D + 15	Hemorrhagic cystitis	NA	PO; 1200 mg/d × 44 d	Vidarabine	Failure	Received 5 days of GCV + vidarabine without effect, then switched to RBV
**SOT Cases:**											
Blohmé [134]	1992	1	28 Y/M	Kidney Tx	D + 25	Hemorrhagic cystitis	B7	IV GCV	Reduction in immunosuppression	Recovery	
Duggan [136]	1997	1	58 Y/F	Heart Tx	5 years from Tx	Pneumonia	NA	IV GCV	IVIG	Recovery	
Lim [141]	2005	1	51 Y/M	Kidney Tx	D + 36	Nephritis	NA	IV GCV followed by PO VGCV	Reduction in immunosuppression	Recovery	
Kozlowski [142]	2011	2	44 Y/M; 56 Y/M	Kidney Tx	A few days before D + 24; D + 19	Nephritis; one pt also had PNA	B34	PO VGCV	Reduction in immunosuppression	Recovery	Donor-derived from the same donor
Nanmoku [143]	2016	6	33–45 Y/ 2 F and 4 M	Kidney Tx	D + 7 to D + 1763	All with hemorrhagic cystitis; three pts also with nephritis and two of these with viremia	NA	IV GCV	Reduction in immunosuppression in three pts	Recovery (all 6)	Two pts also had BK viruria
Paula [144]	2016	1	32 Y/M	Kidney Tx	1 month post Tx	Nephritis + viremia	NA	IV GCV	Reduction in immunosuppression	Recovery	
Barros Silva [145]	2017	1	38 Y/M	Kidney Tx	18 months post Tx	Nephritis + viremia	NA	GCV	IVIG; Reduction in immunosuppression	Recovery	
Moreira [146]	2019	1	40 Y/M	Kidney Tx	D + 17	Nephritis	NA	Iv GCV followed by PO VGCV	IVIG; Reduction in immunosuppression	Recovery	

* Disseminated infection is defined by involvement of two or more end-organs. Involvement of a single end-organ with viremia is not classified as disseminated disease. ** Dosing of IV GCV or oral VGCV were largely based on recommended dosing for CMV infection, with renal adjustment as needed. *** Failure is defined as lack of clinical and/or virologic response to treatment. For those patients with death reported, cause of death was attributed to AdV infection only for some. **Abbreviations:** AdV, adenovirus; Allo-HCT, allogeneic hematopoietic cell transplant; AML, acute myeloid leukemia; Auto-HCT, autologous hematopoietic cell transplant; CMV, cytomegalovirus; GCV, ganciclovir; HCT, hematopoietic cell transplant; IV, IV; IVIG, IV immunoglobulin; MDS, myelodysplastic syndrome; MM, multiple myeloma; MRD, matched related donor; MUD, matched unrelated donor; NA, not available; PO, per os; pt, patient; SOT, solid organ transplant; Tx, transplant; VGCV, valganciclovir.

**Table 7 viruses-17-00599-t007:** List of studies using virus specific T-cells for treatment of AdV infection in HSCT patients.

Author [Reference]	Year	PMID/Clinical Trial No.	Study Description	No. of Patients (Total/with AdV)	Method of CTL Isolation/ Production	Results: Virological and /or Clinical Response	Results: Survival/Mortality Impact	Adverse Events
**Studies using donor-derived VST’s**								
Leen [165]	2009	19700662/ NCR00590083	Clinical trial to assess the safety of cytotoxic T lymphocytes for prevention and treatment of EBV and AdV	12 received for prophylaxis and 1 for AdV Disease	Produced from donor PBMC’s using AdV vector	Two patients with AdV disease cleared the infection; 11 patients who received ppx did not develop disease	/	None
Gerdemann [166]	2013	23783429/ NCT01070797	Phase 1/2 trial to study the safety and efficacy of VST’s for AdV, CMV and EBV	10/5 AdV (*n* = 1) EBV + AdV (*n* = 2) CMV + AdV (*n* = 2)	DNA plasmids to generate donor-derived virus-directed T-cell lines with specificity for AdV, EBV, and CMV	Complete response in all five patients	/	None
Papadopoulou [167]	2014	24964991/ NCT01570283	Clinical trial assessing the feasibility and clinical utility of VSTs against EBV, AdV, CMV, BKV, HHV-6	11 /1	Rapidly-generated single-culture VSTs that recognize 12 immunogenic antigens from five viruses using allogeneic stem cell donor	94% with complete virological response; one AdV infection resolved	Two died due to non-infectious causes	Skin GVHD in one patient
Feucht [168]	2015	25617426/ 2005-001092-35 EU Clinical trial register	Clinical trial to analyze the safety and efficacy of ex vivo adoptive T-cell transfer for AdV	30/30	PBMCs were isolated from stem cell donors, after being stimulated by hexon protein	86% with complete clearance of viremia (Responders)	71% (15) of responders survived; 100% (eight out of eight) non-responders died. Attributable mortality due to AdV is 100% in non-responders and 9.5% responders.	Mild GVHD grade 1 in two patients within 2 weeks after ACT and GVHD grade 2–3 in four patients > 7 weeks after ACT
Creidy [169]	2016	27246524/ NCT01325636	French multicenter pilot trial to treat peds/adult patients post-HSCT with CMV/AdV infection with VST’s	15/8 5 AdV 3 AdV + CMV	Donor cells were stimulated with hexon AdV antigen followed by magnetic enrichment of IFN-γ–secreting cells using the Cytokine secretion system and the CliniMACS device	Of the five patients alive, four showed a complete virological response and one was a no response	Two died prior to 21 day evaluation and one death was attributable to AdV	Four patients with respiratory failure (unclear if it was related to AdV or ACT) and one patient developed GVHD but unclear if associated with ACT
IP, [170]	2018	29753677/ NCT01822093	Open-label phase 1/2 study to assess the safety of pre-emptive administration of AdV-specific T-cells to treatment AdV viremia in high-risk pediatric patients after HSCT.	8/8	AdV-specific T-cells were expanded from donors using peptides and cytokines	All eight patients with complete virological response	Two deaths (one due to AdV infection despite clearance of viremia)	Grade 4 GVHD in one patient
Abraham [171]	2019	31292125/ NCT0880789 NCT01923766	Clinical trial to evaluate the feasibility and safety of CB-derived multivirus-specific T-cells in pediatric patients	14/1	PBMCs from cord blood donors were used to generate LCLs and dendritic cells, then transduced with AdV vector or with peptide mix	All who received CB-VSTs as ppx did not develop any end-organ disease from CMV, EBV, or AdV; one patient with AdV disease resolved	Two died; not attributable to viral infection	Grade 3 GI GVHD in one patient
Rubinstein [172]	2022	35108727/ NCT03883906	Single-arm, phase 2 study to assess the efficacy of donor-derived VST’s in prevention of viral infection due to CMV, AdV, BKV and EBV	23/NA	PBMC’s were stimulated with peptide mixes	21% (five) with treatment failures; three developed significant viremia/viral disease requiring additional antiviral therapy, one due to AdV.	Four deaths; not attributable to viral infection	GVHD in two patients
	**Studies using 3rd-party VST’s**							
Leen [173]	2013	23610374/ NCT00711035	Multicenter study of 3rd-party VSTs to treat CMV, EBV and AdV post-HSCT	50/18	PBMCs were transduced with Ad5f35pp65 vector	6-week cumulative response rate of 77.8% (95% CI, 53.7–100%) for AdV	Five deaths; all attributable to AdV	Two patients with grade 1 GVHD de novo
Tzannou [174]	2017	28783452/ NCT02108522	Phase 2 clinical trial using off the shelf T-cells for treatment of multiple viruses ( CMV, AdV, EBV, BKV, HHV6) post-HSCT	38/7	Posoleucel—3rd-party VSTs generated using peptide multimers	four CR; one PR; two non-response (Cumulative response rate of 71.4%)		Recurrent grade 3 GI GVHD in one patient and grade 1–2 skin GVHD in five patients
Pfieffer [175]	2023	36628536/ NCT02108522	Open-label, phase 2 trial to determine the feasibility and safety of posoleucel in HSCT recipients with AdV, BK virus, CMV, EBV, HHV-6, and JC virus.	58/12	Posoleucel—3rd-party VSTs generated using peptide multimers	6-week response was observed in 10 of 12 patients (83%; 95% CI, 51.6–97.9%),	/	13/58 (22%) patients developed GVHD
Keller [176]	2024	38637498/ NCT03475212	Phase 2 multicenter study using partially-HLA matched VSTs targeting CMV, EBV or AdV	51/30 24 AdV 6 CMV + AdV	/	Virological response in 74% (17/23)	Overall survival 57.1% (95% CI: 42.00–70.00%) at 1 year	Grade III cytokine release syndrome occurred in one patient requiring treatment with tocilizumab and steroids. Graft rejection in one patient associated with infusion
	** Studies using both donor- derived and 3rd-party VST’s **							
Qian [177]	2017	2848908/ NCT0285157	Phase 1/2 multicenter pilot study involving the infusion of AdV-VST after HSCT in the event of refractory ADV infection or disease.	11/11	AdV-VST generated by interferon (IFN)-γ-based immunomagnetic isolation from their original donor (42.9%) or a third-party haploidentical donor (57.1%)	Virological response in 91% patients	four deaths; one attributable to AdV	GVHD in three patients
Rubenstein [55]	2021	34473237/ NCT02048332 NCT02532452	Single-center phase 1/2 clinical trial to assess safety and efficacy of VSTs for treatment of adenoviremia	29/29 7 DD 21 TP 2 both	PBMCs were stimulated with pools of viral peptides (Pepmix) encompassing antigen epitopes.	Clinical response in 81%, with a CR in 58%; CR and overall response rates were higher in patients treated with DD VSTs compared with TP (86% vs. 42% CR; 74% vs. 100% overall response)	/	GVHD in one patient who received 3rd-party VSTs
Galletta [178]	2023	36736781/ NCT02532452 NCT02048332	Retrospective cohort study of patients who received VSTs for treatment of AdV, BKV, CMV and EBV	145/37 77 DD 68 TP	PBMCs were stimulated with pools of viral peptides (Pepmix) encompassing antigen epitopes.	Clinical response rate for AdV was 64.9% . No difference in outcomes between DD and TP	81% survived in DD and 66% in TP group after 1 year	Eight and five patients from DD and TP donors developed GVHD

**Abbreviations:** ACT; adoptive T-cell therapy, AdV; adenovirus, BKV; BK polyomavirus, CB; cord blood, CMV; cytomegalovirus, CR; complete virological response, DD; donor-derived, EBV; Epstein–Barr virus, GI, gastrointestinal; GVHD; graft-versus-host disease, HLA; human leukocyte antigen, HHV-6; human herpesvirus 6, HSCT; hematopoietic stem cell transplant; IFN; interferon, LCL; lymphoblastoid cell line, PBMC; peripheral blood mononuclear cell, ppx; PR, partial virological response; prophylaxis, TP; third party, VSTs: virus specific T-cells.

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
