# Peer review of "Treating Adenovirus Infection in Transplant Populations: Therapeutic Options Beyond Cidofovir?"

_viruses, 2025, doi:10.3390/v17050599_

Round 1
Reviewer 1 Report
Comments and Suggestions for Authors
In their manuscript entitled “Treating Adenovirus Infection in Transplant Populations: Therapeutic Options Beyond Cidofovir?” by Narsana et al. explores established and emerging treatment options for hematopoietic stem cell recipients and solid organ transplant recipients suffering from an adenovirus infection. The focus of this review was on the antiviral agent cidofovir noting its broad activity against human AdV species but emphasizing its treatment limiting factor nephrotoxicity, disqualifying it from being the forefront of treatment options. Alternative antivirals such as brincidofovir, ribavirin, and ganciclovir were considered, citing variable efficacy data spanning pediatric to geriatric populations. Emerging therapies, particularly adoptive T cell therapy, was highlighted and noted to provide promising outcomes with the concern of graft-versus-host disease.
The manuscript effectively presented a comprehensive overview of antiviral treatment options for adenovirus, supported by well-curated literature. The inclusion of both established and emerging therapies, such as adoptive T cell therapy is a particular strength of this review. Additionally, the manuscript's clear structure and logical organization enhance readability and allowed the reader to easily follow the progression of ideas. Overall, the authors succeed in presenting a well-supported and accessible review of current and evolving therapeutic strategies.
A possible limitation to this manuscript may derive from the challenge of formulating suitable guidance as to a “standardized treatment protocol” and an appropriate design for “comparative studies to identify optimal dosing and timing.” Nonetheless, these needs remain and the review highlighted the progress and requirement for further progress in these directions. It may be beyond the scope of this review, but perhaps the authors could have provided a stronger opinion on both of these topics to prioritization future research and therapeutic strategies.
In sum, this review serves and important need and provides a timely and solid review of the state-of-the-art for treating adenovirus disease in the immunocompromised population.
Minor concerns:
- The inclusion of data on complications, efficacy, and age-stratified mortality on alternatives to cidofovir—like that provided for other antivirals—might strengthen this report and facilitate further research.
- The discussion of ribavirin focuses primarily on hematopoietic stem cell recipients, with limited reference to the solid organ transplant population. Is ribavirin used less frequently for solid organ trnsplants? If so, perhaps a comment on why or why not would be helpful.
- In Tables 4, 5, and 6, organizing patient outcomes by age might facilitate clearer comparisons across age groups and enhance the applicability of treatment strategies to specific populations.
Author Response
We would like to express our sincere gratitude to the reviewers for their insightful comments and constructive suggestions, which have significantly enhanced the quality of our manuscript. Please find the detailed responses below and the corresponding revisions/corrections highlighted/in track changes in the re-submitted file
Comment 1:
The inclusion of data on complications, efficacy, and age-stratified mortality on alternatives to cidofovir—like that provided for other antivirals—might strengthen this report and facilitate further research.
Response 1:
Thank you for your suggestions. We have added side effects of ribavirin and ganciclovir based on your suggestion. The efficacy of all these agents has already been discussed. Most of the studies have heterogenous population (including both pediatric and adult populations) which would make it difficult to assess age stratified mortality for these agents.
Additional text has been added from lines 323-338 and 478-482. Additional references 86a – 86f have been added from lines 815-825 and 141a -141b from lines 928-931 in the revised manuscript.
Comment 2:
The discussion of ribavirin focuses primarily on hematopoietic stem cell recipients, with limited reference to the solid organ transplant population. Is ribavirin used less frequently for solid organ transplants? If so, perhaps a comment on why or why not would be helpful.
Response 2:
Thank you for your suggestion.
In our literature search, we identified only 7 cases of adenovirus infection in SOT populations that were treated with ribavirin. All these cases are included in our manuscript. It is possible that HSCT patients are more susceptible to severe adenovirus infections compared to SOT patients, which may explain the discrepancies in the number of cases reported in HSCT versus SOT populations. Due to the paucity of data in SOT populations, it is challenging to draw any meaningful conclusions.
Comment 3:
In Tables 4, 5, and 6, organizing patient outcomes by age might facilitate clearer comparisons across age groups and enhance the applicability of treatment strategies to specific populations.
Response 3:
Thank you for your suggestion.
We currently have the case reports and retrospective studies organized in chronological order. Since most of the retrospective studies involved mixed populations that include both pediatric and adult patients, it would not be feasible categorize the patient’s outcomes by age.
Reviewer 2 Report
Comments and Suggestions for Authors
In this manuscript, Narsana, Ha, and Ho review the current literature on treatments for adenovirus (AdV) infections. These treatments include antivirals and adoptive T cell therapy. The manuscript is thorough, with extensive tables that include multiple studies. At times, the manuscripts is not concise and can jump/not flow in a manner that makes it easy for the reader to follow (an example is in the specific comments). Overall, the manuscript should be accepted after some minor additions/edits
General Comments:
Tables 4, 5, Supplemental Figure 4, and others: The / in the column of Species/Serotype makes it look like it wasn’t filled in. If the paper did not determine which species/serotype, put in the column an abbreviation for did not determine (such as DND) and define the abbreviation in the table legend.
Specific Comments:
Line 38-39: Reword or remove (most viruses are a family/not single virus)
Lines 45-46: The authors mention receptors, but do not specifically mention which ones/based on types. This should be added.
Lines 67-73: This is a jump from the previous paragraph. This should be moved to after line 83
Line 107: needs reference
Lines 287-293: This needs to be rewritten with the summary (no quotes) and unpublished data or personal communication (give the name)
Lines 294-301: References needed
Lines 348-349: This sentence can be removed (this is a review, just state the findings)

Author Response
We would like to express our sincere gratitude to the reviewers for their insightful comments and constructive suggestions, which have significantly enhanced the quality of our manuscript. Please find the detailed responses below and the corresponding revisions/corrections highlighted/in track changes in the re-submitted file
General Comments
Comment 1:
Tables 4, 5, Supplemental Figure 4, and others: The / in the column of Species/Serotype makes it look like it wasn’t filled in. If the paper did not determine which species/serotype, put in the column an abbreviation for did not determine (such as DND) and define the abbreviation in the table legend.
Response 1:
Thank you for your suggestion. We agree with this comment. We have replaced “/” with “NA” (Not Available) in the columns of Species/Serotype of these tables accordingly. We also added the abbreviation in the Table legend.
Specific Comments
Comment 1:
Line 38-39: Reword or remove (most viruses are a family/not single virus)
Response 1:
Thank you for your suggestion. We agree with this comment. We made changes in the text in line 39. The updated version is “It is important to appreciate that human AdV is a family of viruses.”
Comment 2:
Lines 45-46: The authors mention receptors, but do not specifically mention which ones/based on types. This should be added.
Response 2:
Thank you for your suggestion. We agree with this comment. We made changes in the text in line 48-50 in the revised manuscript.
Comment 3:
Lines 67-73: This is a jump from the previous paragraph. This should be moved to after line 83
Response 3:
Thank you for your suggestion. The paragraph from Lines 67-73 discusses the epidemiology of AdV infection, while the 2 paragraphs from lines 76-83 and 84-92 discuss the clinical diseases and outcomes with AdV. Thus, we prefer not to insert the paragraph of Lines 67-73 after line 83.
Comment 4:
Line 107: needs reference
Response 4:
Added the reference per recommendation (line 109 of the revised manuscript).
Comment 5:
Lines 287-293: This needs to be rewritten with the summary (no quotes) and unpublished data or personal communication (give the name)
Response 5:
Thank you for your suggestion. We agree with this comment. We changed the text to reflect a summary rather than quotes. These changes were made in Line 292-296 in the revised manuscript.
Comment 6:
Lines 294-301: References needed
Response 6:
Thank you for your suggestion. We agree with this comment. We added the references.
These changes were made in lines 300, 302 and 305 of the revised manuscript.
Comment 7:
Lines 348-349: This sentence can be removed (this is a review, just state the findings)
Response 7:
Thank you for your suggestion. We agree with this comment. We removed this sentence.
The changes were made in line 367 and 368 of the revised manuscript.